# Cytogenetics of Pediatric Acute Myeloid Leukemia: A Review of the Current Knowledge

**DOI:** 10.3390/genes12060924

**Published:** 2021-06-17

**Authors:** Julie Quessada, Wendy Cuccuini, Paul Saultier, Marie Loosveld, Christine J. Harrison, Marina Lafage-Pochitaloff

**Affiliations:** 1Hematological Cytogenetics Laboratory, Timone Children’s Hospital, Assistance Publique-Hôpitaux de Marseille (APHM), Faculté de Médecine, Aix Marseille University, 13005 Marseille, France; julie.quessada@ap-hm.fr; 2Aix Marseille University, CNRS, INSERM, CIML, 13009 Marseille, France; marie.loosveld@ap-hm.fr; 3Hematological Cytogenetics Laboratory, Saint-Louis Hospital, Assistance Publique des Hôpitaux de Paris (APHP), 75010 Paris, France; wendy.cuccuini@aphp.fr; 4Groupe Francophone de Cytogénétique Hématologique (GFCH), 1 Avenue Claude Vellefaux, 75475 Paris, France; 5APHM, La Timone Children’s Hospital Department of Pediatric Hematology and Oncology, 13005 Marseille, France; paul.saultier@ap-hm.fr; 6Faculté de Médecine, Aix Marseille University, INSERM, INRAe, C2VN, 13005 Marseille, France; 7Hematology Laboratory, Timone Hospital, Assistance Publique-Hôpitaux de Marseille (APHM), 13005 Marseille, France; 8Leukaemia Research Cytogenetics Group Translational and Clinical Research Institute, Newcastle University Centre for Cancer Faculty of Medical Sciences, Newcastle University, Newcastle upon Tyne NE1 7RU, UK; christine.harrison@newcastle.ac.uk

**Keywords:** cytogenetics, pediatrics, acute myeloid leukemia, genomics, risk-adapted therapy, therapeutic trials, children hematological malignancies, karyotype, FISH, acute megakaryoblastic leukemia, infant leukemia

## Abstract

Pediatric acute myeloid leukemia is a rare and heterogeneous disease in relation to morphology, immunophenotyping, germline and somatic cytogenetic and genetic abnormalities. Over recent decades, outcomes have greatly improved, although survival rates remain around 70% and the relapse rate is high, at around 30%. Cytogenetics is an important factor for diagnosis and indication of prognosis. The main cytogenetic abnormalities are referenced in the current WHO classification of acute myeloid leukemia, where there is an indication for risk-adapted therapy. The aim of this article is to provide an updated review of cytogenetics in pediatric AML, describing well-known WHO entities, as well as new subgroups and germline mutations with therapeutic implications. We describe the main chromosomal abnormalities, their frequency according to age and AML subtypes, and their prognostic relevance within current therapeutic protocols. We focus on de novo AML and on cytogenetic diagnosis, including the practical difficulties encountered, based on the most recent hematological and cytogenetic recommendations.

## 1. Introduction

Acute leukemia (AL) is the most frequent cancer in children. The majority of cases comprise acute lymphoblastic leukemia (ALL), whereas only 15–20% have a diagnosis of acute myeloid leukemia (AML). Pediatric AML is thus a rare disease, with an incidence of seven cases per million children younger than 15 years, affecting children with a median age of 6 years [1,2]. Children have better outcomes than adults because of the more frequent presence of good prognostic genetic features and higher tolerance of intensive treatment. Complete remission (CR) is now achieved in 90% of cases, whereas event-free (EFS) and overall survival (OS) rates are commonly around 50% and 70%, respectively, due to the high rate of relapse. Moreover, short and long term therapy-related toxicities have to be taken into consideration, with a persistent high risk of death due to intensive therapy (4–10%) and significant long term side-effects of certain chemotherapies (anthracycline) [1,2,3,4,5,6,7,8].

Although in adults a large proportion of AML cases are secondary to a previous myelodysplasia (MDS) or previous exposure to radio- or chemotherapy (therapy-related AML), in children, 95% of cases represent de novo AML. However, rare cases of pediatric AML have an underlying constitutional genetic predisposition either in the context of a phenotypically apparent syndrome, such as Down syndrome, or as a more subtle familial syndrome [9,10]. AML in these inherited situations may be preceded or not by a myelodysplastic syndrome. Thus, detection is crucial in order to adapt the treatment, because some may be sensitive to chemo- or radiotherapy. Additionally, the risk of a bone marrow allograft with an affected family member needs to be avoided [10].

Chromosomal abnormalities are recognized as an important factor in diagnosis and as an independent prognostic indicator. AML cells (blast cells) are malignant myeloid progenitor cells that fail to differentiate, proliferating in the bone marrow and invading peripheral blood and other organs, such as the central nervous system. Clonal, acquired, somatic cytogenetic abnormalities (CAs) are detected in 75 to 80% of pediatric AML cases [1]. They are either primary, detected in all cytogenetically abnormal cells, or secondary, being present in one or multiple subclones, indicating clonal evolution. Primary CAs are closely associated with morphological subtypes. In 1976, the French–American–British (FAB) cooperative group proposed an AML classification based on morphological and cytochemical criteria, dependent on cell lineage and the degree of differentiation; for example, in the revised 1985 FAB classification, acute myelomonocytic leukemia with abnormal eosinophils, M4eo [11,12]. FAB classification has been regularly updated and continues to be used, because it is available on the day of diagnosis, which may assist in the search for a specific FAB subtype CA, such as the inversion of chromosome 16, inv(16)(p13q22), in a M4Eo AML [11,12]. At the end of the 1990s, the World Health Organisation (WHO) proposed a consensual extended classification based on clinical, morphological, immunophenotypical, cytogenetic and molecular characteristics, although the majority of the categories remained closely aligned to the FAB subgroups [13,14,15]. The WHO classification is regularly updated, taking into consideration those molecular abnormalities resulting from the cytogenetic abnormalities; for example, AML M4Eo/inv(16)(p13q22) is now referred to as AML with inv(16)(p13q22);*CBFB-MYH11* [14]. Such molecular abnormalities, immunophenotypic features and recurrent mutations found in AML provide powerful markers for the detection of minimal (measurable) residual disease (MRD), which is used as a prognostic marker in current AML treatment trials [2]. Furthermore, the CD33 antigen characteristic of AML is also a target for immunotherapy in some current AML trials [16,17,18].

One significant AML subtype, acute promyelocytic leukemia (APL), M3-M3v/t(15;17)(q24;q21), now referred to as APL with *PML-RARA*, can be cured by treatment based on a vitamin A derivative (ATRA) and arsenic trioxide (ATO), within specific APL protocols [19,20].

Initial risk stratifications for treatment in most AML therapeutic protocols are based on chromosomal and molecular abnormalities of the leukemic cells. Overall, three prognostic categories are distinguished: good (favorable), intermediate and adverse (poor). The definition of standard-risk may be misleading because it is applied either to favorable (non-high-risk) or to intermediate risk groups, dependent on the therapeutic protocol [2].

In terms of clinical impact, adult classification systems, such as those defined by ELN, cannot be completely transferred to the classification of childhood AML, because the cytogenetic (and genomic) landscapes of pediatric and adult AML and the cytogenetic risk associations are different according to age [1,8,21,22,23]. 

The most frequent cytogenetic abnormalities in pediatric AML are balanced chromosomal rearrangements, leading to the formation of chimeric fusion genes, only some of which are also found in adults [24]. Among them, core binding factor (CBF) leukemias, represented by t(8;21)(q22;q22)/*RUNX1-RUNX1T1* and inv(16)(p13q22)/*CBFB-MYH11*, are associated with a good prognosis, whereas 11q23/*KMT2A* (*MLL*) rearrangements are associated with an intermediate or adverse prognosis, depending on the *KMT2A* partner gene involved. In rare cases, such as inv(3)(q21q26), the balanced rearrangement leads to a positional effect with the relocation of enhancer sequences to the vicinity of a protooncogene, whose expression becomes upregulated [25,26]. Unbalanced abnormalities, such as monosomies of chromosomes 5 and 7, are less frequent in children, but they are associated with a poor outcome, as observed in adults [1].

Age is an important factor among children, as indicated by the age-specific FAB/WHO subtypes. Infant AML was historically defined as AML occurring in children under 1 year of age, but it has now been extended to include all children under 2 years, because they share the same clinical and biological profiles [27]. For example, they include a higher proportion of acute megakaryoblastic leukemia (AMKL, M7), whereas certain cytogenetic abnormalities are exclusively identified in this age group, such as t(1;22)(p13;q13)/*RBM15-MKL1* [1,4,22,28]. 

Cytogenetic analysis complemented by FISH (see graphical abstract) continues to play an important role in the diagnosis of AML, providing a rapid, global genome analysis, using FISH for the detection of cryptic chromosomal rearrangements, such as t(5;11)(q35;p15)/*NUP98-NSD1* [29,30,31,32,33]. In addition, classical chromosomal rearrangements can appear as variant “masked” forms, involving a third chromosome or as a cryptic insertion, requiring confirmation by FISH [34]. Alternatively, the detection of chimeric fusion transcripts by reverse transcriptase polymerase chain reaction (RT-PCR) or real-time quantitative PCR (RQ-PCR) may routinely be used. Furthermore, RQ-PCR enables the subsequent detection of MRD with strong clinical value in current therapeutic protocols [1,2]. 

Other molecular cytogenetic techniques, such as CGH or SNP-array, are used for the detection of unbalanced cytogenetic abnormalities (gain or loss of material) [35]. Finally, the detection of mutations of prognosis significance, such as *FLT3-ITD*, *NPM1* and *CEBPA*, particularly in cases with normal karyotypes, is mandatory in current therapeutic protocols [2,36].

Here, we provide an updated review of cytogenetic abnormalities in pediatric AML. We describe the main chromosomal rearrangements (balanced and unbalanced, primary and secondary), their frequency according to age and AML subtypes, and their prognostic significance within current therapeutic protocols. We focus on de novo AML and cytogenetic diagnosis based on the most recent hematological and cytogenetic recommendations [1,2,31,32,37,38].

## 2. Cytogenetic Subgroups

Among recurrent CAs we can considerer balanced CAs, like translocations and inversions, and unbalanced CAs, like monosomies, trisomies or deletions (Table 1, Figure 1).

### 2.1. Balanced Cytogenetic Abnormalities

Recurrent balanced genomic rearrangements are predominant in pediatric AML. Downstream consequences of these rearrangements may be: (1) the formation of a fusion gene, which encodes an oncogenic chimeric protein, or (2) more rarely, the overexpression of a gene implicated in self-renewal, cellular cycle or another major cellular function, by relocation to the vicinity of enhancer sequences (Table 1).

#### 2.1.1. Acute Promyelocytic Leukemia (APL), M3-M3v/t(15;17)(q24;q21), Now Referred to as APL with *PML-RARA*

Acute promyelocytic leukemia (APL) represents around 5 to 10% of childhood AML, although the frequency varies between countries and ethnicities, being more prevalent in Hispanic populations. APL is very rare in infants, with a median age in children of about 12 years [1,2,72,73]. Most cases occur de novo, although therapy-related cases, arising secondary to exposure to alkylating agents, topoisomerase 2 inhibitors, or radiation, have been described mainly in adults but also in children [74]. APL represents the M3/M3v FAB AML subtype and is characterized by the presence of the *PML-RARA* fusion gene, classically due to the translocation, t(15;17)(q24;q21). The resulting chimeric protein induces a blockade in granulocytic differentiation at the promyelocytic stage [75]. Essentially, it is an aggressive leukemia with a high rate of disseminated intravascular coagulation (DIVC), possibly leading to fatal hemorrhages; thus, APL constitutes an emergency at diagnosis, even with low WBC counts [1,20]. However, the existence of an effective targeted therapy, applied in a timely manner, changes it to a good risk AML subtype. Indeed, all-trans retinoic acid (ATRA) and/or ATO induce granulocyte terminal differentiation and clinical remission. APL cases are now included in specific APL protocols with a very high cure rate [19,20,76,77]. Of note, 5–10% of APL/*PML-RARA* cases present with a cryptic insertion, ins(15;17) or ins(17;15), leading to the formation of the *PML-RARA* fusion gene. In these cases, FISH, using appropriate probes covering the 5′ part of PML and the 3′ part of RARA, is very useful for rapid emergency diagnosis [34]. On the other hand, RT-PCR or RQ-PCR is always run in parallel in M3/M3v cases and, in practice, will confirm cytogenetic results by demonstrating the presence of a *PML-RARA* transcript. Furthermore, RQ-PCR enables subsequent quantification of MRD [20]. Additional chromosomal abnormalities (ACA) are found in 30% of APL at diagnosis, the most frequent being trisomy 8 or the gain of 8q, del(9q) and ider(17)(q10)t(15;17) leading to *TP53*/17p deletion. The prognostic value of these ACAs is unclear. A recent study comprising pediatric and adult cases treated with ATRA and chemotherapy found a higher risk of relapse in cases with more than two ACAs [78], a feature reminiscent of the less favorable prognosis observed in such cases in an adult ATO-based trial [79]. As commented in the same issue by Grimwade and Dillon [80], the prognostic relevance in cases with complex karyotypes could be related to the high frequency of 17p deletions at the karyotype level in this adult series. Similarly, the prognosis value of *FLT3-ITD* or *FLT3* mutations, present in about half of APL cases, remains unknown in the context of ATRA and/or ATO-based therapies, which may be correlated with a WBC higher than 10 × 10^9^/L, which is a well-established poor risk prognosis factor [20,34,73,80,81,82]. 

Notably, AML cases with *RARA* partners other than *PML* have been described. These cases present with a cytological profile similar to AML M3/M3v: AML with *NPM1*/t(5;17)(q35;q21), *NUMA1*/t(11;17)(q13;q21), *PLZF*/t(11;17)(q23;q21), *PRKAR1A*/del(17)(q21;q24)/t(17;17)(q21;q24), *FIP1L1*/t(4;17)(q12;q21), *BCOR*/t(X;17)(p11;q12), der(17) and t(3;17)(p25;q21). It is important to identify these very rare cases of “APL-like” leukemia because some of them, such as those involving *PLZF*, are resistant to treatment with ATRA or ATO, thus warranting a standard AML protocol [73,81].

#### 2.1.2. Core Binding Factor AML 

Acute myeloid leukemia with t(8;21)(q22;q22)/*RUNX1-RUNX1T1* and inv(16)(p13q22) or t(16;16)(p13;q22)/*CBFB-MYH11* are referred to as core binding factor (CBF) leukemia. They represent the largest pediatric AML subgroup, accounting for around 25% of cases, being rare in infants, both with a median age of 8–9 years among children [1,27]. They are detected at a lower frequency among young adult AML [83]. Both rearrangements lead to the disruption of CBF genes involved in the CBF complex, which plays a major role in hematopoiesis. Both fusion genes provide excellent markers for molecular MRD monitoring. 

Both rearrangements block myeloid differentiation, but alone they are insufficient to induce overt leukemia. As a result, additional cytogenetic abnormalities (ACA) and/or somatic mutations are present in all cases, with a median of seven mutations per case, mostly seen in those genes activating tyrosine kinase signaling such as *KIT, N/KRAS* and *FLT3* (mainly *FLT3-ITD* and *-TKD* mutations) [40,41,84,85,86]. In contrast to inv(16) AML, t(8;21) AML presents with a high frequency of mutations in genes regulating chromatin conformation (epigenetic modifiers) such as *ASXL1* and *ASXL2* or encoding members of the cohesin complex such as *RAD21* and *SMC1A* [41,42].

Altogether, CBF leukemias are associated with a relatively good prognosis with an overall survival rate >80%, although the incidence of relapse remains around 30% [1,16,22,28,36].

The translocation t(8;21)(q22;q22) results in the fusion of 5’sequences of *RUNX1* (21q22) and 3’ sequences *RUNX1T1* (8q22), giving rise to the *RUNX1-RUNX1T1* functional chimeric gene [85]. It occurs in 12–15% of childhood AML, and in 80% of cases it is associated with the FAB classification AML M2 subtype [1,16,22,28,36]. Typically, blasts present with single and thin Auer bodies and dysgranulopoiesis. Moreover, the phenotype frequently shows aberrant expression of CD19 and CD56, allowing accurate MRD monitoring [87]. In the largest international study comprising 835 t(8;21) pediatric patients, ACAs were found in 68% of cases: loss of X or Y chromosome (46%), del(9q) (12%), trisomy 8 (6%), abnormal(7q) (5%), trisomy 4 (3%), and, in 25% of cases, at least two of these ACA were present [40]. These abnormalities have been confirmed and common deleted regions (CDRs) were refined by CGH or SNP array analyses, e.g., del(7)(q35q36.1) and del(9)(q21.2q21.3) for abnormal 7q and del(9q), respectively [35,43]. Classically, the presence of these ACAs does not modify the good prognosis associated with t(8;21) AML [22]. However, in the largest international retrospective study, del(9q) has been significantly associated with a lower CR rate, without impacting EFS and OS, whereas trisomy 4 cases were significantly associated with a higher cumulative incidence of relapse (CIR) and lower EFS and OS [40]. Of note, the *KIT* gene is located at 4q11 and trisomy 4 cases are associated with a higher rate of *KIT* mutations [40]. Moreover, a high *KIT* variant allele frequency (VAF) and co-occurrence of tyrosine kinase pathway mutations with mutations in epigenetic modifying or cohesin genes have been associated with a higher CIR in t(8;21) AML, and thus a less favorable prognosis [42].

As described in other types of AML, cryptic cases occur in around 10% of AML/*RUNX1-RUNX1T1* as variant translocations involving a third chromosome or as cryptic insertions confirmed by accurate FISH detection [88]. 

The inv(16)(p13q22) and the rare translocation t(16;16)(p13;q22) result in the fusion of 5’sequences of *CBFB* (16q22) and 3’ sequences of *MYH11* (16p13), leading to a *CBFB-MYH11* functional chimeric gene [84]. They account for 7–11% of pediatric AML and are typically associated with the AML M4 with abnormal eosinophils (M4 Eo) FAB subtype [1,16,22,28,36]. Thus, a (myelo)monocytic leukemia with eosinophils presenting with purple granulations is an important pointer to this rearrangement even though it may also occur in classical AML M4 or AML M5 subtypes. This abnormality is sometimes difficult to identify by karyotyping; thus, specific FISH and/or PCR is warranted in order to confirm its presence, especially in M4 Eo or cases with trisomy 22. Indeed, trisomy 22 is a characteristic and frequent ACA, other frequent ACA include del(7q) and trisomy 8 [43]. CGH array analyses have refined the del(7q) CDR as del(7)(q35q36.1) [35,43].

The prognostic value of ACA in CBF leukemia is still debated. Trisomy 8, which is twice more frequent in inv(16) than in t(8;21), has recently been found as the genetic aberration with the strongest negative impact on prognosis in a large adult CBF AML study [89], confirming the previous results in inv(16) CBF adult AML [90]. Altogether, CBF leukemia remains a favorable risk group and the prognostic value of secondary genetic abnormalities within this group warrants confirmation on large prospective therapeutic trials [2]. 

#### 2.1.3. *KMT2A*/11q23 Rearrangements

Rearrangements involving *KMT2A* (*KMT2A*r, previously mixed lineage leukemia, *MLL*) gene located at 11q23.3, account for 15–20% of pediatric AML [1,44]. They are more frequently associated with monocytic AML (FAB M4 or M5 in 73% of *KMT2A*r cases), but can also be found in M0 (3%), M1 (6%) and M7 (3%) subtypes [44]. *KMT2A*r AML are more frequent in infant AML, accounting for 47–55% of children under 2 years of age; thus, the median age in childhood AML is 2.2 years [27,44].

*KMT2A* encodes a protein of the nuclear structure involved in the regulation of transcription and epigenetic modulations. For each rearrangement, the 5’ part of *KMT2A* fuses with the 3’ part of a partner gene, leading to a chimeric functional gene located, in classical translocations, on the derivative chromosome 11, der(11). *KMT2A*r is also found in ALL, and, between both types of acute leukemia, more than one hundred *KMT2A* partners have been identified [45]. *MLLT3(AF9)* (9p21.3) is the most common partner, representing 46% of all pediatric *KMT2A*r AML. It is the only *KMT2A*r AML identified in the current WHO classification, as AML with t(9;11)(p21.3;q23.3); *KMT2A*-*MLLT3* [14]. This subtype is associated with an intermediate prognosis, as is also the case for t(11;19)(q23;p13) either with *ELL* (19p13.1) or *MLLT1(ENL)* (19p13.3) partners, which account for 8% and 6% of *KMT2A*r cases, respectively. Other *KMT2A*r partners are rare, such as *MLLT6 (AF17)* (17q21) and *SEPT6* (Xq24) [44]. Exceptionally, t(1;11)(q21;q23); *KMT2A-MLLT11(AF1Q)* AML has been associated with a good prognosis [44]. However, due to the scarcity of these cases, this association has not been confirmed [22,28,36]. 

In contrast, a poor prognosis has been assigned to *KMT2A*r cases involving *MLLT10 (AF10)* (10p12), *ABI1* (10p11.2) and *MLLT4 (AF6)* (6q27), accounting for 16%, 2% and 6% of *KMT2A*r cases, respectively [1,44]. Moreover, the fusion partner distribution is variable from one age group to another. *MLLT10* (10p12) and *ABI1* (10p11.2) are prevalent in infant AML, whereas *MLLT4* (6q27) is more frequent in older children (median age 12 years) [44]. 

In the largest international study to date, additional cytogenetic abnormalities (ACAs) were found in about half of *KMT2A*r AML (47%), mainly as chromosomal gains, trisomy 8 being most prevalent (18% of total cases) followed by trisomies 19, 6 and 21, each found in 5% of total cases. Cases with at least two ACAs accounted for 26% of cases. This study confirmed the prognostic value of the different *KMT2A*r partner genes and found that trisomies 8 and 19 were predictive of a better and a worse prognosis, respectively [58]. Unbalanced CAs have been confirmed by SNP array analyses [35].

*KMT2A*r AML present with a low mutation burden and a characteristic gene expression and methylation profile [46]. Interestingly, the EVI1 protooncogene is overexpressed in about 40% of *KMT2A*r AML, and this overexpression may worsen the prognosis of *KMT2A/**MLLT3* cases [50,91]. 

The numerous *KMT2A* partners indicates the use of a *KMT2A* break-apart FISH probe for initial diagnosis, complemented by specific probes for the identification of those partner genes with an important prognosis impact. Multiplex PCR may be troublesome for use in diagnosis, because sometimes breakpoints require cloning/sequencing [45]. Indeed, these *KMT2A*r abnormalities may be difficult to detect by karyotyping, especially those involving *MLLT10* (10p12), because this gene is oriented telomere to centromere. Thus, in order to create a functional chimeric *KMT2A*-*MLLT10*, a simple reciprocal translocation between chromosomes 10 and 11 is not sufficient, and other mechanisms, such as inversion or insertion, are necessary to correctly orientate the gene segments. Thus, accurate FISH analysis, using a break-apart probe covering *KMT2A* and/or a fusion probe covering both genes, is recommended alongside PCR analyses for accurate diagnosis of this poor prognosis rearrangement [92]. Moreover, t(10;11)(p12;q23)/*KMT2A-MLLT10* can closely resemble t(10;11)(p12;q14)/*PICALM/MLLT10*, a rare abnormality currently assigned to an intermediate risk group [54]. FISH analysis has revealed that t(10;11)(p12;q14)/*KMT2A-MLLT10* are in fact complex rearrangements, such as der(10)t(10;11)(p12;q14)inv(11)(q14q23), in which the *KMT2A* gene is disrupted by an 11q inversion on the derivative 10 chromosome, in order to produce an in-frame *KMT2A-MLLT10* fusion. 

In fact, even in classical t(v;11q23)/*KMT2A* translocations, FISH diagnosis may be difficult, because the 3’part of the *KMT2A* gene, which is usually translocated to the partner chromosome, can be deleted in about 10% of cases [93]. Finally, some partner genes have been identified on the 11q chromosome such as the *CBL* gene located at 11q23.3, telomeric of *KMT2A*, and in such cryptic deletion, del(11)(q23.3q23.3) cases, FISH analysis provides evidence of deletion of the 3’part of the *KMT2A* gene [45,94]. 

#### 2.1.4. 11p15/*NUP98* Rearrangements

The 11p15 rearrangements involving nucleoporin 98Kd (*NUP98*) are rare, occurring in 3–5% of pediatric AML and rare cases in young adults [36,47,50,51,95,96]. They are a relatively biological and clinical homogeneous group with a poor prognosis which can be overcome by allogeneic hematopoietic stem cell transplantation (HSCT) [95]. This poor risk is mainly due to a high rate of induction failure [97]. 

Multiple *NUP98* partners have been described, but the most frequent is the nuclear receptor-binding SET domain protein 1 (*NSD1*) gene (5q35), accounting for about 75% of *NUP98*r pediatric cases. The chimeric protein, resulting from the fusion between the 5’part of *NUP98* and the 3’part of *NSD1*, induces the self-renewal of myeloid stem cells and enhances the expression of HOX genes [98]. This translocation, t(5;11)(q35;p15), must systematically be screened for because it is a cryptic cytogenetic abnormality [29]. It is found in 8 to 16% of pediatric AML, with apparently normal karyotype, but it may be associated with non-specific CA, such as trisomy 8 [47,51,95]. *NUP98*-*NSD1* leukemias are frequently associated with *FLT3* -ITD and/or *WT1* mutations, which occur in about 80% and 50% of cases, respectively, possibly adding to the poor prognosis associated with these cases [36,49,95].

Among other *NUP98* partners, the most frequent is *KDM5A*
*(JARID1A)* at 12p13.3, leading to an almost-cryptic cytogenetic abnormality, initially described in M7 pediatric AML) [52]. *NUP98*-*KDM5A* occurs in 2% of pediatric AML, mainly in the M7 subtype, and is a poor prognosis marker with an overall survival rate of around 33% [96,99]. *NUP98*-*KDM5A* accounts for 9–12% pediatric M7 and 12% of infant AML [27,48,63]. In contrast to *NUP98-NSD1*, few additional mutations are found in association with *NUP98*-*KDM5A*, suggesting that the fusion protein itself has a sufficient oncogenic effect [99]. Finally, *NUP98* rearrangements occur with a high frequency, in around one-third of cases, with the rare acute erythroid leukemia M6 subtype [99,100]. 

FISH, using a *NUP98* break apart probe, is now widely used for diagnoses of *NUP98*r, and metaphase FISH enables identification of the partner gene [47,95,101]. Of note, because commercially available *NUP98* probes are 5′ (upstream) and 3′ (downstream) flanking probes, one has to be cautious in the interpretation of interphase FISH results and rather allow a larger distance between 5′ and 3′ signals rather than a two-spot distance between these signals.

#### 2.1.5. 12p Abnormalities Including the Rare t(7;12)(q36;p13)/*ETV6*;*MNX1*


Abnormalities of the short arm of chromosome 12 (12p) and, more particularly, those involving the *KDM5A* located at 12p13.3 (as a *NUP98* partner described above) and the ets variant 6 gene (*ETV6*), at 12p13.1, are found in 4% of cases and are associated with an adverse prognosis [22,28]. 

The rare subtle (often cryptic translocation) t(7;12)(q36;p13) presents with a breakpoint 5′ of *ETV6* and a variable breakpoint of proximal 7q (upstream) to *MNX1* (7q36.3) (for a review, see Espersen et al., 2018 [53]). It induces ectopic expression of the *MNX1* (*HLXB9)* homeobox transcription factor with the blockade of differentiation and senescence in hematopoietic progenitors and stem cells. Indeed, an *ETV6-MNX1* transcript has never been found, and the reciprocal *MNX1-ETV6* is only observed in 50% of cases [102,103]. Therefore, FISH analysis provides the most powerful tool for diagnosis. However, due to the wide variability of 7q breakpoints, the existence of deletions on the derivative 7q, three-way translocations and cryptic insertions, and the choice of accurate FISH probes is crucial [53,104,105]. To date, t(7;12)(q36;p13)/*ETV6-MNX1* has only been described in infants (under 2 years old) with an incidence of 4.3% in infants and 1.1% in pediatric AML, as reported in a recent review [53]. ACAs are present in 86% of cases, and all cases with ACAs had trisomy 19 [53]. In the literature, a high rate of relapse has been reported (77%); however, the salvage rate using HSCT is high [53]. Therefore, FISH screening for this poor prognostic abnormality should be mandatory in infants under 2 years old, especially in cases with trisomy 19 [4,31,38,105].

### 2.2. Rare Balanced Rearrangements 

#### 2.2.1. Inversion (3;3)(q21q26.2)/t(3;3)(q21;q26.2)/*GATA2*;*MECOM (EVI1)*

These abnormalities, sharing the same chromosomal breakpoints, are included in the WHO 2016 classification as inversion, inv(3)(q21q26.2)/translocation, t(3;3)(q21;q26.2) with involvement of *GATA2* (3q21), and *MECOM (EVI1)* (3q26.2). Both rearrangements result in repositioning of a distal enhancer of *GATA2* to the vicinity of *MECOM*, thus resulting in the overexpression of *MECOM* and silencing of *GATA2* [25,26]. This abnormality is well described in adult cases and characterized by an unusual normal or high platelet count, dysmorphic platelets and megakaryocytes, with monosomy 7 as a frequent secondary cytogenetic abnormality [106]. This poor prognostic abnormality occurs at a low frequency (1–2%) in childhood AML, with a median age of 3 years [1,22,24]. An association between central diabetes insipidus and AML with inv(3)(q21q26) and/or monosomy 7 has rarely been described in children and adults; most cases present with both of these CAs [5,107]. The pathophysiological mechanisms underlying central diabetes insipidus in these patients remain unclear, but could be related to abnormal platelets because most of the peripheral ADH is platelet-bound; most of these cases have a documented response to a vasopressin analog [107].

#### 2.2.2. Translocation (6;9)(p22;q34)/*DEK-NUP214*

The translocation t(6;9)(p22;q34) is a rare rearrangement occurring in 1–2% of childhood AML cases, mainly in older children with a median age of 12 years, with no infant cases reported [1,57,108]. It leads to the fusion of the 5’part of the *DEK* gene located at 6p22, encoding for a nuclear phosphoprotein, and the 3′ part of a nucleoporin gene, *NUP214* (*CAN)* located at 9q34 [109]. It has been considered as a distinct entity of the WHO classification since 2008. Classically, this abnormality is mainly in found in M2 or M4 AML FAB subtypes, all presenting with myelodysplasic features, and in one-third of cases with mild basophilia in the bone marrow [57,108]. ACAs are present in 19% of cases: mainly, a loss of the Y chromosome and trisomies 8 and 13 [108]. *FLT3-ITD* is present in 40–70% of cases. This abnormality is associated with a poor prognosis with high risk of initial treatment resistance and high risk of relapse (CIR 57–64%). This poor outcome persists independent of the presence of *FLT3-ITD*, but it may be improved by HSCT in the first CR [57,108]. 

#### 2.2.3. Translocation t(3;5)(q25;q35)/*NPM1-MLF1*

Translocation t(3;5)(q25;q35), mainly described in young adults, is a rare entity (about 0.5% of AML) identified in AML with myelodysplastic features and M6 cases [110,111,112]. It produces a fusion of 5’ coding sequences of the nucleophosmin (*NPM1)* gene at 5q35 and the myelodysplasia/myeloid leukemia factor 1 (*MLF1*) gene on 3q25, producing an *NPM1-MLF1* in-frame chimeric gene [113]. Cases in children are extremely rare: a recent review of the literature collected eight pediatric AML patients with a median age of 3.5 years (range 2–13). Most patients were M2 or M4 with only one M6 case, mostly with no ACAs [55]. Of note, only 3/8 cases were confirmed by FISH and/or RT-PCR. This scarcity impairs the assignment of a precise prognosis value to these cases that are currently classified as intermediate risk at diagnosis. 

#### 2.2.4. Translocation t(8;16)(p11;p13)/*KAT6A-CREBBP*


The translocation t(8;16)(p11;p13) is a rare entity leading to the fusion of the histone acetyltranferase gene *KAT6A* (*MYST3* or *MOZ*) at 8p11 with the *CREBBP* (*CBP*) gene at 16p13 [114]. In an international study, which collected 62 pediatric cases, the median age was 1.2 years (range 0–16 years) with a high frequency of neonates (one-third of patients were younger than 1 month old). Most cases were M4-M5 FAB subtype with a high rate of hemophagocytosis, leukemia cutis, and disseminated intravascular coagulation (DIVC). About half of cases presented with ACAs, but only a few were recurrent: trisomy 1q, del(9q), trisomy 8, del(5q) and del(7q). No difference in prognosis was found when compared to an unselected pediatric AML cohort. Interestingly, about one-third of neonates experienced a spontaneous remission, and half of them remained in continuous remission [58,115]. 

#### 2.2.5. t(16;21)(p11;q22)/*FUS-ERG*

The t(16;21)(p11;q22) leads to the in-frame fusion of the 5′ part of the *FUS* gene (16p11) and the 3′ part of the *ETS* related gene, *ERG* (21q22) [116,117]. A recent international collaborative study has collected 31 cases of this rare abnormality [59]. These cases represented 0.5% of the COG AAML31 cohort and 0.3% of the BFM cohort. There were no infant cases; the median age was 8.5 years (range 2.0–17.5 years) with no specific FAB subtype. Most cases were de novo AML (2/31 were t-AML), and the prognosis was poor, with a CIR of 74% and a 4-year EFS of 7% (15% for allografted cases). ACAs were present in 71% of cases, mainly described as “complex” karyotypes with at least two ACAs (32%); trisomy 8 (19%) and, unexpectedly, trisomy 10 (13%) was prevalent. 

#### 2.2.6. t(16;21)(q24;q22)/*RUNX1-CBFA2T3*

The rare t(16;21)(q24;q22) leads to the in-frame fusion of the 5′ part of the *RUNX1* gene (21q22) and the 3′ part of the *CBFA2T3* gene (16q24.3) [118]. The same international collaborative study mentioned above collected 23 cases [59]. These cases represented 0.3% of the COG AAML31 and 0.1% of the BFM cohorts. The median age was 6.8 years (range 1.0–17 years) and M1 and M2 FAB subtypes were significantly prevalent (76%). Of note, as reported in adults but to a lesser extent, therapy-related cases were observed (22% of cases). Overall, the outcomes were good, with a CIR of 0% and a 4-year EFS of 77%. ACAs were present in 84% of cases, with trisomy 8 (42%) and loss of the Y chromosome (43% of male patients) being prevalent. Their gene expression profile was closely related to that the t(8;21)(q22;q22)*RUNX1/RUNX1T1* cases which share the same 5′ *RUNX1* part of the fusion gene [59].

#### 2.2.7. Translocation (1;22)(p13;q13)/*RBM15-MLK1*

The translocation t(1;22)(p13;q13) is a very rare abnormality (0.3% of pediatric AML) included in the WHO classification. It is only seen in infants and toddlers (median age 0.7 years, range 0.1–2.7 years) and in AMKL cases (5 to 10% of non-DS-AMKL) (Table 2 and Figure 2) [22,48,61,62,63,64]. It leads to fusion of the 5′ part of *RBM15* (OTT) and the 3′ part of *MKL1* (*MAL*) located at 1p13.3 and 22q13.2, respectively. In a knock-in murine model, this fusion induces abnormal megakaryopoiesis and transformation to AMKL, similar to the human form of the disease, with hepatosplenomegaly and liver and bone marrow fibrosis [48,119,120]. This entity shows an intermediate outcome [48,63]. Of note, a high proportion of normal metaphases are seen in the karyotypes, which present mainly with few ACAs (in fewer than one-third of cases and mainly in older infants), typically as hyperdiploid karyotypes with duplication of the der(1)t(1;22), and gains in chromosomes 2, 6, 19, and 21 [48,60,61,64]. Furthermore, the frequently associated myelofibrosis can impair cytogenetic and PCR analyses; thus, FISH screening for this primary abnormality provides an appropriate diagnostic test in infant AMKL [64]. 

#### 2.2.8. The Cryptic Inversion, inv(16)(p13.3q24.3)/*CBFA2T3-GLIS2*

In 2012, a cryptic inversion of chromosome 16, inv(16)(p13.3q24.3)/*CBFA2T3-GLIS2*, was identified in 27–31% of non-DS pediatric AMKL, thus representing the most frequent aberration found in de novo pediatric AMKL (Table 2 and Figure 2) [121,122]. This abnormality results in fusion of the 5’ part of *CBFA2T3* (*ETO2*) (16q24) and the 3’ part of *GLIS2* (16p13.3), leading to an increase in self-renewal capacities of megakaryoblastic progenitors by the upregulation of ERG and downregulation of GATA1. [123] Later, this abnormality was shown to not be restricted to AMKL [66], although half of reported cases of *CBFA2T3-GLIS2* AML cases were AMKL, with a median age of 1.5 years (range 0.5–4 years), a female predominance (two-thirds of cases), and a poor prognosis [48,63,64]. This poor prognosis is also shared by non AMKL *CBFA2T3-GLIS2* cases, who are usually older children of median age 12.4 years (range 0.3–17.2 years). In a study focusing on normal karyotype (NK) pediatric AML, *CBFA2T3-GLIS2* AML accounted for 8% of NK-AML, whereas another study reported that these cases represented 2% of patients with *FLT3-ITD* at a low allelic ratio [66,67]. Of note, two-thirds of *CBFA2T3-GLIS2* AML cases presented with ACAs, mostly as chromosomal gains leading to hyperdiploid karyotypes (47–49 chromosomes), with trisomy 3 being characteristic and present in 20% of cases, followed by trisomy 21 and gain of the Y chromosome [48,67]. Interestingly, *CBFA2T3-GLIS2* AML presents with a peculiar “RAM” immunophenotype characterized by high CD56 (NCAM) expression and low or no expression of HLA-DR, CD45 and CD38 antigens. This aberrant RAM phenotype can assist in diagnosis, MRD monitoring, and providing the possibility of targetable anti CD56 therapy [67,124]. These patients showed a poor outcome, with only half achieving complete remission; 25% presented with extramedullary disease and overall survival rates ranging from 15 to 30% [48,64,65,66,67]. 

### 2.3. Unbalanced Cytogenetic Abnormalities

In addition to classical reciprocal translocations, other types of recurrent cytogenetic abnormalities occur in pediatric AML, including the gain or loss of material or numerical aberrations, which are found in around 40% of childhood AML [56]. The most prognostically significant are monosomy 5, deletion of the long arm of chromosome 5 (del(5q)), and monosomy 7. Although they are associated with a poor outcome, they occur in fewer than 5% of patients [1,18,22,28]. 

#### 2.3.1. Partial or Total Loss of Chromosomes

##### Monosomy 7 and del(7q)

Monosomy 7 and deletion of the long arm of chromosome 7, del(7q), are frequent in childhood myelodysplasic syndromes (MDS), where they account for 40% cases [125]. In pediatric AML, in the largest retrospective study published to date, these abnormalities were compared [69]. Both abnormalities may be present as secondary abnormalities. Deletions (7q), are more frequently associated with CBF leukemia, whereas monosomy 7 is more often associated with adverse primary abnormalities, such as inv(3)(q21q26) described above. Nevertheless, when they are considered as sole abnormalities, monosomy 7 and del(7q) occur at a similar frequency of 3%, and occur at a similar median age (7.2 years and 7.6 years, respectively). However, monosomy 7 cases have a poor prognosis (5-year OS, 35%, 5-year EFS 28%) whereas in del(7q) patients, prognosis is intermediate (5-year OS, 43%, 5-year EFS 39%) [22,28,69]. In more recent studies, the poor prognosis of monosomy 7 was confirmed to be mainly due to a higher risk of resistance to induction therapy (71%–83% CR) [22,28]. Of note, both abnormalities, and especially monosomy 7, can be acquired during the evolution of predisposition syndromes, such as GATA2 deficiency and *SAMD*9/*SAMD9L* germline mutations syndromes, although can also be found in apparent “de novo” AML [126,127].

##### Monosomy 5 and del(5q) (-5/5q-)

Thus far, in the largest pediatric AML study, 26 cases of -5/5q- among 2240 cases were identified (1.2%). Median age was 12.5 years (0.3–20.7 years) and two-thirds of patients were over 11 years of age [68]. A significant association with the FAB M0 subtype was found (24%). No cases had monosomy 5 as the sole abnormality, and most patients (81%) presented with ACAs, mainly as complex karyotypes: two-thirds with at least two ACAs and half of cases with at least three ACAs. Among these ACAs, loss of 17p, identified by karyotyping, was prevalent (23% of all cases) mostly among complex karyotypes. Of note, only two cases presented with del(7q), and no cases had monosomy 7. As reported in adults, the prognosis was poor, with a 5-year EFS of 23% and an overall survival (OS) of 7%, similar to that previously reported by the UK MRC trial [22]. However, in this study, the authors noted that it was difficult to assign a poor prognosis to del(5q) alone, because these cases are very rare. Additionally, in this study, complex karyotypes were not found to show prognostic significance. Of note, del(5q) occurs as a secondary abnormality in at least two subtypes of AML with cryptic abnormalities: t(5;11)(q35;p15)/*NUP98-NSD1* and the rare t(7;21)(p22;q22)/*RUNX1-USP42*, thus emphasizing the need for a complete cytogenetic and molecular screening of such cases [29,128,129].

#### 2.3.2. Gains of Chromosomes 

##### Trisomy 8

Trisomy 8 is found in around 10 to 14% of childhood AML, either as the sole cytogenetic change or associated with another structural of numerical abnormality (see below, hyperdiploid karyotypes) [22,70]. Trisomy 8, as the sole abnormality, is found in only 3% of cases. It is more frequently associated with older age in children (median age 10.1 years) and *FLT3-ITD* mutations [70]. In the most recent BFM trial, trisomy 8 as a sole abnormality had a poor outcome, but no molecular data were provided. Notably, trisomy 8 occurs mainly as a secondary cytogenetic change, thus indicating the need to search for a primary cryptic abnormality, such as 11p15/*NUP98r* [5].

#### 2.3.3. Complex, Hyperdiploid and Monosomal Karyotypes

Complex karyotypes (CKs) and monosomal karyotypes (MKs) are well known poor prognosis factors in adult AML [111,130]. However, there is no consensus in pediatric AML. In the BFM98 trial analysis, CK, defined as “three or more CAs, including at least one structural CA, excluding favorable cytogenetics and *KMT2Ar*”, was a poor risk factor found in 8% of cases [28]. However, in the MRC trial, using a similar definition (*KMT2Ar* were not excluded), CKs were represented in 15% of cases, and showed an intermediate prognosis. Even if CK was expanded to include at least five CAs, or if CK was divided into typical complex karyotypes (comprising -5/5q-, monosomy 7 or del(17p)and atypical CK, without these abnormalities, no association with poor risk emerged [22]. 

In relation to CK, three or more numerical gains, without structural abnormalities, would also be considered as CK. In the first large study of such hyperdiploid cases, with 49–65 chromosomes seen at the karyotype level in AML, only two children with AML M7 were included (1 DS et 1 non-DS) [131]. Two subsequent large studies, including pediatric cases, identified that, in hyperdiploid cases with solely numerical gains, these gains were not random, with trisomies 8, 21, 19 and 6 being most frequently observed. These cases, without accompanying adverse cytogenetic rearrangements, were not shown to have the poor prognosis normally considered for CK [56,71]. Hyperdiploidy with a modal number between 49 and 65 chromosomes, was associated with infant cases, acute megakaryoblastic leukemia, and lower WBC [56].

Monosomal karyotypes (MKs) have been defined as two or more autosomal monosomies or one autosomal monosomy with at least one structural abnormality (excluding marker or ring chromosomes) and without the favorable CA: (t(15;17)(q22;q21); t(8;21)(q22;q22); inv(16)(p13q22)/t(16;16)(p13;q22)) [130]. In the more recent BFM trial, using the same previous BFM criteria for CK definition, CK cases had a poor prognosis only if they were monosomal. Furthermore, all MK cases (3% of total cases) were associated with a poor prognosis, even after the exclusion of monosomy 7. MK was associated with a younger age (median age 3.9 years) and showed significantly lower EFS and OS (23% and 35%, respectively) compared to other patients. This poor prognosis of MK was worsened in hypodiploid karyotypes [5]. The NOPHO-DBH-AML study confirmed the poor prognosis of MK (5-year EFS 34% vs. 49%, for non-MK cases) with more frequent refractory disease, although the OS was similar to non-MK patients [6].

### 2.4. Normal Karyotypes 

Normal karyotypes account for around one-quarter (22–26%) of pediatric AML [22,28,36]. Their risk assignment is based on the search for cryptic cytogenetic abnormalities (for example, *NUP98*r) and for somatic mutations with well-established prognosis relevance, such as *NPM1*, *FLT3-ITD* and *CEBPAdm*, in the current adult and pediatric therapeutic trials [7,8,132]. Indeed, even though these mutations are found at a much lower level in children than in adults, increasing with age, they share the same prognostic significance [132,133]. *FMS*-like tyrosine kinase 3-internal tandem duplication (*FLT3-ITD*), found in 10–20% of pediatric AML, was initially associated with a poor prognosis if the mutated allele ratio (ITD-AR, ITD to wild-type ratio) was high [134]. However, updated analysis has suggested that even at a lower ratio (0.1–0.4), *FLT3-ITD* retains its poor prognostic impact [135]. These patients may benefit from a targeted therapy in the same way as adult cases [136]. Notably, co-occurrence of *NPM1* mutations, associated with a good prognosis, overrides the poor prognosis associated with *FLT3-ITD* [46,137]. Conversely, *WT1* mutation co-occurrence with *FLT3-ITD* worsens the prognosis [46,135]. Of note, as emphasized in the adult ELN 2017 risk classification (see Table 3), *FLT3-ITD* mutations should not be used as adverse prognostic markers if they occur within favorable cytogenetic risk groups such as CBF leukemia.

*NPM1* and *CEBPA* double mutations (dms), accounting for around 9% and 4% of pediatric AML cases, respectively, are associated with a good prognosis. They are mainly found in normal karyotype cases, assigning them to a low risk category [36,137,138,139]. Of note, *CEBPA* may be a germline mutation, predisposing to AML, especially in cases with a double mutation (dm), emphasizing the need for investigation of these patients in remission. If one of the mutated alleles remains at a high variant allele frequency (VAF), of around 50%, suspicion of an inherited mutation is high, highlighting the need for genetic counseling in order to confirm the constitutional and familial origin of this mutation [10,140].

Large studies using high throughput sequencing have confirmed these mutations, well-known fusion genes, and copy number alterations (CNAs), but also have identified novel mutations and fusion genes, highlighting age-specific cytogenetic/molecular profiles and precising the molecular landscape of pediatric AML [4,46,50,96].

**Table 2 genes-12-00924-t002:** Main cytogenetic subgroups in pediatric AMKL.

Cytogenetic Subgroups	Fusion Gene or Genes Involved	Frequency in Non-DS AMKL	Median Age, (Range), years	Special Features	Secondary CA	Secondary Molecular Abnormalities	Prognosis	References
DS AMKL								
Trisomy 21c	*GATA1* (Xp11) truncating mutation	NA	1.7 (0.4–3.8)	85–97% of DS-AML were M7 TAM in 25% of DS pts that can evolve towards M7 in 10% of cases	tri 8, gain of a third chr 21, gain of 1q	Mutations in cohesin complex genes *(STAG2, RAD21, …)*, *MPL, RAS, JAK2, JAK3*	Good (impaired by trisomy 8?)	[9,141,142,143,144]
Non-DS AMKL			1.6 (0.1–17)	Mainly infants Hepatosplenomegaly Myelofibrosis that can impair sampling for diagnosis				[48,62,63,64]
inv(16)(p13.3q24.3) *	*CBFA2T3-GLIS2*	20% (16–27%)	1.5 (0.5–4)	Infants, extramedullary disease, CD56++	tri 21 tri 3	Low frequency of mutations	Very Poor	[48,63,64,65]
t(1;22)(p13;q13)	*RBM15-MKL1*	12–14%	0.7 (0.1–2.7)	Only M7 Hepatosplenomegaly, Fibrosis	Mainly no ACA HD karyotypes, tri 1q (unbalanced t(1;22) in 26% of cases)	Low frequency of mutations	Intermediate	[48,60,61,62,63,64]
11q23.3/*KMT2*r	*KMT2A with multiple partners*	10–15%	1.9 (0.7–12)	Only 3% of *KMT2A*r pediatric AML were M7	tri 19, tri 21	Low frequency of mutations, overexpression of HOX genes	Poor	[44,48,62,63,64]
t(9;11)(p22;q23)	*KMT2A-MLLT3 (AF9)*	6–10%						
t(10;11)(p12;q23)/ ins(10;11)(p12;q23q13) ***	*KMT2A-MLLT10 (AF10)*	1–3%						
t(6;11)(q27;q23)	*KMT2A-MLLT4 (AF6)*	1%						
t(11;17)(q23;q12)	*KMT2A-MLLT6 (AF17)*	0.7–1%						
t(11;19)(q23;p13.3)	*KMT2A-MLLT1 (ENL)*	0.5–1%						
t(4;11)(q21;q23)	*KMT2A-AFF1 (AF4)*	0.5%						
12p13 abnormalities	*NUP98-KMD5A* *ETV6 (12p13.1)* del(12p)						Poor	[22,28]
t(11;12)(p15;p13) *	*NUP98-KMD5A*	10%	1.9 (0.8–8.5)	34% of cases were M7	CK (numerous numerical and structural CA); *RB1* deletion (13q14)	Low frequency of mutations; low RB1 expression; overexpression of HOX genes	Poor	[47,48,52,63,64]
t(7;12)(q36;p13) *	*ETV6*; *MNX1*	very rare	0.5 (0.2–1.9)	4/42 cases were M7 Only infants	tri 19 (3/4 cases)	Unknown	Poor	[53]
*HOX-r*	*HOX family genes (HOXA9, HOXA10, HOXB9, …)*	14%			trisomy 19, trisomy 21	Overexpression of HOX genes	Good	[64,65]
t(3;7)(q21;p15.2)	*GATA2-HOXA9*	rare						[64]
t(3;7)(q21;p15.2)	*GATA2-HOXA10*	rare						[64]
t(5;7)(p13.2;p15.2)	*NIPBL-HOXA9*	rare						[64]
t(5;17)(p13.2;q21.3)	*NIPBL-HOXB9*	rare						[64]
t(11;22)(q24;q12)	*MN1-FLI1*	rare						[141]
*GATA1* mutation	*GATA1* (Xp11) truncating mutation	7%		Search for a DS (mosaicism)	tri 21 in nearly all cases	Same gene expression profile as DS-AMKL	Good	[64]
Monosomy 7	/	7–8%	1.5 (0.5–17.1)	Exclude a primary abnormality and a predisposition syndrome *(GATA2)*	/	Frequently as part of a complex karyotype	Poor	[48,62,63,64]
Abnormal 7p	unknown *(HOXA9?)*	12%	1.8 (0.5–8.2)	50% of abn7p cases were translocations; search for *HOX*r (7p15)			Good? Intermediate?	[62,64]
del13q	unknown *(RB1?)*	4%	1.5 (0.6–4.9)	Search for a primary stratifying CA that can be cryptic (*NUP98-KDM5A*)				[62]
Hyperdiploidy (47–84 chr)	% in AMKL: tri 21 (36%), tri 19 (24%) tri 8 (20%) tri 6 (15%)	50%		Search for a primary stratifying CA that can be cryptic	/	/	According to cryptic CA and mutations or intermediate	[48,62,63]
Hyperdiploidy (47–50 chr)	/	38%	1.7 (0.1–15)	Search for a primary stratifying CA that can be cryptic	/	/		[62]
Hyperdiploidy (51–84 chr)	/	12%	1.7 (0.6–6.5)	Search for a primary stratifying CA that can be cryptic	/	/		[62]
Complex	At least 3 independent CAs including a structural CA	50%	1.5 (0.4–15)	Search for a primary stratifying CA that can be cryptic	/	/	According to cryptic CA and mutations or intermediate	[5,6,22,28]
Normal karyotype	/	13–16%	1.5 (0.1–16)	Search for a cryptic CA or prognostic mutation	/	/	According to cryptic CA and mutations or intermediate	[48,62,63]

Abbreviations: AMKL: acute megakaryoblastic leukemia; CA: cytogenetic abnormality; CK: complex karyotype (at least 3 CAs); HD: hyperdiploidy; mon: monosomy; NA: not applicable; r: rearrangement; TAM transient abnormal myelopoiesis; tri: trisomy. * Cryptic abnormality. ** Infants: children under 2 years old. *** A complex rearrangement or a cryptic insertion is necessary to create a fusion gene (see text).

**Table 3 genes-12-00924-t003:** Pediatric and Adult AML cytogenomic risk stratification according to and adapted from Creutzig et al. [1] and Döhner et al. [21], respectively.

Risk Category	Pediatric AML Risk Stratification	Adult AML Risk Stratification (Excluding APL *)
Favorable	t(15;17)(q24;q21)/*PML-RARA ** t(8;21)(q22;q22)/*RUNX1-RUNX1T1* inv(16)(p13q22) or t(16;16)(p13q22)/*CBFB-MYH11* t(1;11)(q21;q23)/*KMT2A-MLLT11(AF1Q) *** Cytogenetically normal cases with: -*NPM1* mutation; - *CEBPA* double mutation *GATA1* mutation **.	t(8;21)(q22;q22)/*RUNX1-RUNX1T1* inv(16)(p13q22) or t(16;16)(p13q22)/*CBFB-MYH11* *NPM1* mutation without *FLT3-ITD* or with *FLT3-ITD*^low^ † *CEBPA* double mutation
Intermediate	CAs not classified as favorable or adverse	CAs not classified as favorable or adverse t(9;11)(p21;q23)/*KMT2A-MLLT3 (AF9)* ‡ *NPM1* mutation with *FLT3-ITD*^high^ † Wild-type *NPM1* without *FLT3-ITD* or with *FLT3-ITD*^low^ † (without adverse-risk genetic lesions)
Adverse	inv(3)(q21q26) or t(3;3)(q21;q26)/*GATA2*; *MECOM (EVI1)* del(5q), -5 -7 ƒ t(6;9)(p23;q34)/*DEK-NUP214* t(4;11)(q27;q23)/*KMT2A-MLLT2(AF4)* t(6;11)(q27;q23)/*KMT2A-MLLT4(AF6)* t(10;11)(p13;q23)/*KMT2A-MLLT10(AF10)* t(5;11)(q35;p13)/*NUP98-NSD1 *** t(7;12)(q36;p13)/*ETV6(TEL)*; *HLXB9(MNX1) *** t(9;22)(q34;q11)/*BCR-ABL1* Complex karyotype (≥3 CAs) ƒ *FLT3-ITD* mutation § *WT1* mutation §	inv(3)(q21q26) or t(3;3)(q21;q26)/*GATA2*; *MECOM (EVI1)* del(5q), -5 -7 ƒ t(6;9)(p23;q34)/*DEK-NUP214* t(v;11q23)/*KMT2Ar* †† t(9;22)(q34;q11)/*BCR-ABL1* Complex karyotype (≥3 CAs) ƒ -17/abn17p and /or *TP53* mutation # *** Monosomal karyotype ƒƒ *FLT3-ITD*^high^ † § *ASXL1* mutation § *RUNX1* mutation §

NOTE. Favorable, Intermediate, and Adverse were defined according to the definitions given by Creutzig et al., 2012 (1): Favorable indicates 5-year survival >60% in adults and >70% in children; Intermediate, 23–60% in adults and 50–70% in children; and Adverse, <23% in adults and <50% in children. Abbreviations: APL: acute promyelocytic leukemia; CA: cytogenetic abnormality. * t(15;17)(q24;q21)/*PML-RARA* APL is treated separately from other AMLs (see text). ** Abnormalities are rare or absent in adult AML. *** Abnormalities are rare or absent in pediatric AML. † Low, low allelic ratio (<0.5); high, high allelic ratio (≥0.5). ‡ The presence of t(9;11)(p21.3;q23.3) takes precedence over rare, concurrent adverse-risk gene mutations. ƒ In the absence of the WHO-designated recurring translocations or inversions. †† excluding t(9;11)(p21;q23)/*KMT2A-MLLT3.* ƒƒ Defined by the presence of 1 single monosomy (excluding the loss of X or Y) in association with at least 1 additional monosomy or structural chromosome abnormality (excluding core-binding factor AML). # *TP53* mutations are significantly associated with AML with complex and monosomal karyotype in adults. § These markers should not be used as an adverse prognostic marker if they co-occur with favorable-risk AML subtypes.

## 3. Special Considerations: FAB Subtype (M7), Age, Predisposition 

### 3.1. Acute Megakaryoblastic Leukemia 

Acute megakaryoblastic leukemia (AMKL), (FAB classification AML M7), constitutes a distinct AML subtype in children. The accumulation of malignant megakaryoblasts is often accompanied by bone marrow fibrosis that can impair sampling for diagnosis [141]. Two types of AMKL must be distinguished: Down syndrome (DS) and non-Down syndrome AMKL. (Table 2, Figure 2)

Down syndrome (DS) children have a 150-fold higher risk of AML compared to non-DS children, and AMKL is the most frequent AML subtype. It is characterized by a founding *GATA1* mutation, leading to a transient abnormal myelopoiesis (TAM) found in about 25% of newborns, which can evolve to a full AMKL in 10% of cases before the age of 5 years [9,141,142]. DS-AMKL blasts harbor megakaryoblastic and erythoid markers; therefore, it is reported in the WHO classification as “myeloid leukemia associated with Down syndrome” (ML-DS), which shows excellent response and cure with reduced doses of chemotherapy. Of note, acquired cytogenetic abnormalities, present in around two-thirds of cases (mainly trisomy 8, gain of another chromosome 21 and 1q gain), did not impact on the outcome in one study, whereas in another, trisomy 8 indicated a poor prognosis [143,144]. 

Non DS-AMKL occurs in about 10% of pediatric AML, mainly in infants with a median age of about 1.5 years [62,63,64,145,146]. It is globally associated with adverse outcomes, with an overall survival between 42 and 49% and a relapse rate approaching 50% [62,63]. However, as mentioned above, several cytogenetic subgroups are associated with different prognoses. In a recent international collaborative study gathering 153 AMKL cases, t(11;12)/*NUP98-KDM5A*, inv(16)/*CBFA2T3/GLIS2*, *KMT2Ar* and monosomy 7 cases (9%, 16%, 9% and 6% of cases, respectively), defined as the NCK-7 group accounting for 40% of cases, independently predicted a poor outcome. In comparison, the other group including t(1;22)/*RBM15-MKL1*, hyperdiploidy, 7p abnormalities and normal karyotype, accounting for 12%, 22%, 9% and 13% of cases, respectively, showed an improved outcome (4-year OS of 35% vs. 70%, 4-year EFS of 33% vs. 62%, 4-year CIR of 42% vs. 19% for the NCK-7 and the other group, respectively) [64]. Of note, the previously good prognosis associated with 7p abnormalities [62] was not confirmed. Interestingly, 14% of non DS-AMKL are *HOXr* cases, and some of these 7p abnormalities are induced by the rearrangement of *HOXA9* and *HOXA10* genes both located at 7p15.2 [63]. Finally, *GATA1* mutations, which are characteristic of DS-AMKL, can also be found in 9% of non DS-AMKL; trisomy 21 is a constant feature in these cases, raising the possibility of constitutional mosaicism for trisomy 21, as demonstrated in 1/10 of these cases with available non-hematopoietic tissue [63]. Of note, similar findings have been reported in TAM occurring in phenotypically normal newborns, and thus suggests a search for constitutional trisomy 21 mosaicism in these cases [147].

### 3.2. Changes in Cytogenetic and Molecular Genetics According to Age 

A large German study based on routine diagnostic cytogenetic and molecular data in children with AML (more than one thousand pediatric cases (0–18 years) and four thousand adults) confirmed the variation in cytogenetic/molecular subtypes in relation to age between adults and children and among children. Children were separated into three age groups: 0–2 years as infants, 2–12 years and 12–18 years, representing 23%, 40% and 23% of childhood cases, respectively, and striking genomic differences were found. Infants presented with fewer cases in the favorable cytogenetic groups, such as t(8;21) CBF leukemia and t(15;17) APL, with no *NPM1* or *CEBPA* dm cases, whereas *KMTAr* cases were prevalent, mainly with t(9;11)/*KMT2A-MLLT3*. In the 2–12 year range, CBF leukemia was prevalent but slowly decreased thereafter, whereas *KMTAr* cases and especially t(9;11)/*KMT2A-MLLT3* decreased in incidence. Normal karyotypes increased with age from 14% in infants to 27% in older children. The rate of intermediate risk cytogenetic subtypes was similar in all three age groups, at around 50% [24]. 

More recently, a Japanese study analyzed 723 pediatric patients and confirmed the similar genomic profile of children 0–1 and 1–2 years old, including inv(16)/*CBFA2T3-GLIS2* and t(11;12)/*NUP98-KDM5A* cryptic rearrangements. They suggested that, at the genomic and clinical level, cases in children could be separated by a 3-year age threshold, where the *KMT2r* cases fared better and the inv(16)*/CBFB-MYH11* had a less favorable outcome in the younger children. The higher rate of t(9;11) cases in younger children was not retained, because these *KMTAr* subtypes fared better in younger patients, probably due to a significantly lower incidence of high *EVI-1* expression. The inferior inv(16)*/CBFB-MYH11* prognosis seen in infants could be due to the lowered level of intensity of chemotherapy because the increased sensitivity to chemotherapy of younger infants (age < 1 year) was taken into consideration [96].

High-throughput genomic analyses of nearly one thousand pediatric cases, enrolled in successive COG protocols, confirmed a similar genomic profile and indicated the same 3-year threshold. It confirmed the inferior prognosis of *EVI-1* high expression in *KMT2A-MLLT3* patients. These differences in genomic profiles according to age, with an increasing rate of ACA or mutations, likely have some pathogenetic explanations. For example, the presence of fusion genes has been shown in cord blood and differences in the latency period towards the development of AML according to genetic subtypes were seen. Fetal hematopoiesis is retained until the age of 3 years, more recently demonstrated in murine models [23]. However, the 3-year threshold is not a consensus, because other studies have alternatively suggested a 2-year threshold [27,36,148].

Comparison between the most recent international children and adult genomic risk-stratifications at diagnsosis [1,21] (Table 3) shows that most of the AML genetic subgroups are common and share the same prognosis value; for instance, t(8;21)(q22;q22)/*RUNX1-RUNX1T1* and t(6;11)(q27;q23)/*KMT2A-MLLT4* are classified within the favorable and adverse risk subgroups, respectively, whatever the age of the patient. On the other hand, some genetic subgroups with poor prognosis value do not appear in the latest ELN 2017 adult classification, either because they are never observed in adults, such as the infant-specific t(7;12)(q36;p13)/*ETV6*;*MNX1* [53], or are observed in only 2% of adult cases such as the t(5;11)(p15;q35)/*NUP98-NSD1* [95]. Conversely, poor prognostic abnormalities, such as typical complex karyotypes (comprising -5/5q-, monosomy 7 or del(17p)), are rarely observed in children, explaining, at least partly, the discussed value of complex karyotypes in children. In the same way, the scarcity of chromosome 17p abnormalities and *TP53* mutations in children, as recently reported in a large genome sequencing study comparing adult and children mutational landscapes, explains the lack of risk assignment of these abnormalities in children [46]. Interestingly, in the same study, the poor prognosis value of *FLT3-ITD* could be modified by the co-occurrence of *NPM1* mutation, but this fact was observed only in children, probably due to the co-occurrence of *DNMT3A* mutation in adults [46]. 

### 3.3. AML Predisposition Syndromes 

In adults, AML may classically be an evolution of a myelodysplastic or myeloproliferative neoplasm (MDS and MPN, respectively), whereas for children, AML occurring after an MDS/MPN, such as juvenile myelomonocytic leukemia (JMML), are rare [149]. On the other hand, children with constitutional genetic pathologies, such as Down syndrome or inherited bone marrow failure syndrome (IBMFS), have a higher risk of developing AML [9,150]. In the last decade, more subtle phenotypic syndromes linked to germline mutations, conferring a high susceptibility to development of MDS and AML, have been identified [4,10]. Some of them involve genes implicated in normal and malignant hematopoiesis, such as *RUNX1* or *CEBPA.* Taking into account that the same mutations can be acquired in sporadic leukemia emphasizes the need for confirmation of the acquired/constitutional nature of the mutation [151,152]. More recently, *GATA2* has been described as one of the most frequent germline mutated genes predisposing to pediatric AML and MDS, identified with a high frequency (72%) in adolescent AML/MDS with monosomy 7 [153]. Defects in genes implicated in megakaryopoiesis, such as *ETV6* or *ANKRD26*, are known to be responsible for familial thrombopenias and platelet disorders preceding AML, known as FPD/AML [39,154]. Other MDS/AML predisposing genes have been described in the literature, either in the context of a familial history of cancers or bleeding or immunodeficiency syndromes [10,140]. The presence of a germline mutation implicates major consequences for the patient and their family. Thus, every newly diagnosed patient of AML should be screened for potential germline abnormalities, which becomes even more important if these familial hematological disorders are identified in light of potential intrafamilial donors for HSCT [10].

## 4. Cytogenetics Versus Molecular Analysis

Cytogenetic and molecular (cytogenomic) evaluation remains an important part of the diagnosis and prediction of prognosis in pediatric AML, leading to the rapid and accurate assignment of patients to risk-adapted therapies. In adult AML, lack of cytogenetic information (karyotype and subsequent FISH/molecular analyses not performed at diagnosis or karyotype failure) could impair the outcome of these patients because they could not benefit from a risk-adapted therapy [155]. The proposed workflow for pediatric AML at diagnosis (Graphical Abstract) summarizes the laboratory practice according to the most recent recommendations [1,31,32,33,38] and to the current risk-adapted therapeutic trials such as the French-U.K. MyeChild 01 trial (https://clinicaltrials.gov/ct2/show/NCT02724163) accessed on 4 June 2021.

New technologies such as whole genome sequencing (WGS) could replace such workflows in the future, as suggested in a recent cost/effect comparative adult AML study [156]. However, a very high rate of unsuccessful karyotypes was observed in this study (especially if we include karyotypes with no abnormalities and fewer than 20 metaphases analyzed), much higher than the 4% pediatric and adult AML rate observed in the French report of quality indicators (C. Lefebvre and B. Gaillard for the GFCH, manuscript in preparation), the 2–7% rate reported in adult AML [130,155], or the 3–7% range reported in pediatric AML [22,157]. Furthermore, FISH analyses directed by karyotype results, as suggested in our proposed workflow, such as *CBFB-MYH11* fusion probe in cases with 16q22 breakpoint or *KMT2A* break-apart probe in cases with non-informative karyotypes were not applied (but rather applied after WGS), thus limiting the value of this comparative study. On the other hand, this study emphasizes the possibility of reporting genomic results in one week and lowering the current cost of WGS. Most karyotypes and complementary relevant FISH performed for AML at diagnosis are reported within one week, and if we apply a workflow similar to the one proposed here, which relies on current practice according to European/French recommendations, much time and money can be saved.

## 5. Conclusions/Prospective Considerations

Cytogenetics (karyotype and FISH) completed by PCR-based methods and targeted sequencing for the detection of fusion transcripts and mutations remains the gold standard at diagnosis for AML. This combination enables a risk-assignment of nearly each AML case, giving the best chance for the patient to benefit from a tailored therapy.

Moreover, children with de novo AML respond better than adults to intensive therapy and we note continual improvements in outcome over time, in parallel with a better understanding of the cytogenetic and molecular AML landscape and the increasing possibilities of targeted therapies.

## Figures and Tables

**Figure 1 genes-12-00924-f001:**
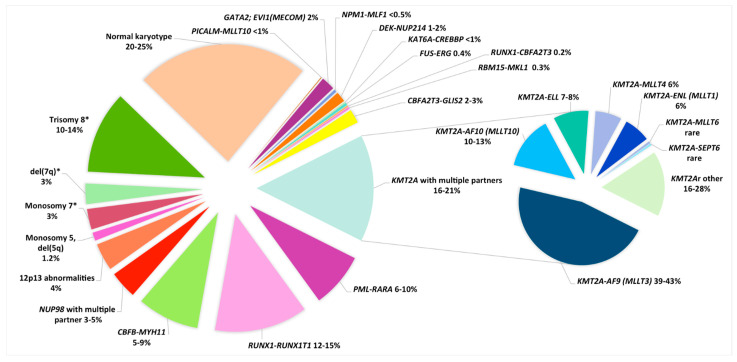
Distribution of cytogenetic subgroups in pediatric AML. * As a sole abnormality.

**Figure 2 genes-12-00924-f002:**
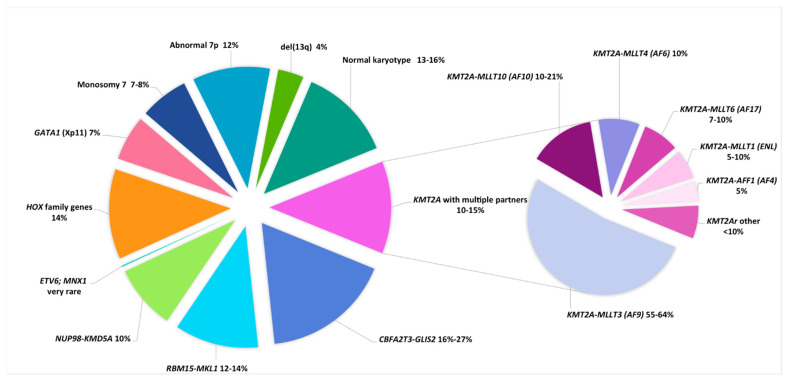
Distribution of cytogenetic subgroups in non-DS pediatric AMKL (adapted from De Rooij 2017 [63] and Masetti 2019 [65].

**Table 1 genes-12-00924-t001:** Main cytogenetic subgroups in pediatric AML.

Cytogenetic Subgroups	Fusion Gene or Genes Involved	Frequency in Childhood AML	Median Age (Y) (Range)	Special Features (Age, FAB, Phenotype, Treatment)	Secondary CA	Secondary Molecular Abnormalities	Risk Category	References
BALANCED CA								
APL								
t(15;17)(q24;q21)	*PML-RARA*	6–10%	12 (1–18)	M3 and M3v, Emergency (DIVC), Specific APL treatment (ATRA, ATO)	tri 8, del(9q), ider(17)(q10)	*FLT3-ITD*	Favorable	[20,39]
CBF leukemias		20–25%						
t(8;21)(q22;q22)	*RUNX1-RUNX1T1*	12–15%	8	M2, blasts with single and thin Auer rods, dysgranulopoiesis, CD19+, CD56+	loss of X or Y, del(9q), tri 8, del(7q), tri 4	*KIT, RAS, FLT3-ITD, FLT3-TKD, ASXL1/2, RAD21*	Favorable	[1,40,41,42,43]
inv(16)(p13q22)/ t(16;16)(p13;q22)	*CBFB-MYH11*	5–9%	9	M4eo	tri 22, del(7q), tri 8	*KIT, RAS, FLT3-TKD, FLT3-ITD*	Favorable	[1,40,41,42,43]
11q23/*KMT2A*r	*KMT2A* with multiple partners	16–21%	2.2 (0–18)	M4 and M5, infants	tri 8	High *EVI1* expression, few mutations	Adverse or Intermediate	[44,45,46]
t(9;11)(p22;q23)	*KMT2A-AF9(MLLT3)*	6–9%	2.6				Intermediate	[44,45,46]
t(11;19)(q23;p13.1)	*KMT2A-ELL*	1–2%	4.6				Intermediate	[44,45,46]
t(11;19)(q23;p13.3)	*KMT2A-ENL(MLLT1)*	1%	7.1				Intermediate	[44,45,46]
t(10;11)(p12;q23)/ ins(10;11)(p12;q23q13) *	*KMT2A-AF10(MLLT10)* *	2–3%	1.3				Adverse	[44,45,46]
t(6;11)(q27;q23)	*KMT2A-AF6(MLLT4)*	1–2%	12.4				Adverse	[44,45,46]
11p15/*NUP98*r	*NUP98* with multiple partners	3–5%	11 (1.3–18)				Adverse	[36,47,48]
t(5;11)(q35;p15) **	*NUP98-NSD1*	3–4%	10.4 (1.2–19.4)	M4,M5 71–77% of *NUP98*r 10–16% of NK	tri 8, del(5q), CK	*FLT3-ITD, WT1*mut	Adverse	[36,46,47,49,50,51]
t(11;12)(p15;p13) **	*NUP98-KMD5A*	1–2%	3.2 (0.01–18.5)	10–30% of *NUP98*r 34% M7, 10% of M7	CK (numerous numerical and structural CA)	Low frequency of mutations	Adverse	[47,48,52]
12p13 abnormalities	*NUP98-KMD5A* del(12p) *ETV6 (12p13.1)*	4%					Adverse	[22,28]
t(7;12)(q36;p13) **	*ETV6*; *MNX1*	1%	0.5 y (0.2–2.3)	Only infants (4% of infants)	tri 19	unknown	Adverse	[53]
Rare other balanced CA								
t(10;11)(p12;q14)	*PICALM-MLLT10*	<1%	older children	Extramedullary disease, granulocytic sarcoma, CD7+	tri 4, tri 19		Intermediate	[46,50,54]
inv(3)(q21q26.2)/ t(3;3)(q21;q26.2)	*GATA2*; *EVI1(MECOM)*	2%	3 (2–18)	Dysmyelopoiesis and platelet abnormalities	mon 7		Adverse	[1,22,24]
t(3;5)(q25;q35)	*NPM1-MLF1*	<0.5%	3.5 (2–13)	M2, M4, M6, dysplasia	rare	unknown	Intermediate	[46,50,55]
t(6;9)(p22;q34)	*DEK-NUP214*	1–2%	12 (2.6–20.4)	M2/M4, dysplasia, basophilia. No infant cases	loss of Y, tri 8, tri 13	*FLT3-ITD*	Adverse	[56,57]
t(8;16)(p11;p13)	*KAT6A-CREBBP*	<1%	1.2 (0–16)	Peak in infants, spontaneous remission in a subset of neonates, DIVC, M4–M5, erythrophagocytosis	tri 1q, del(5q), del(7q), del(9q)	High *HOXA9/HOXA10* expression	Intermediate	[50,58]
t(16;21)(p11;q22)	*FUS-ERG*	0.4%	8.5 (2.0–17.5)	no	tri 8, tri 10		Adverse	[50,59]
t(16;21)(q24;q22)	*RUNX1-CBFA2T3*	0.2%	6.8 (1.0–17)	M1/M2, t-AML	tri 8, loss of Y	Gene expression profile close to *RUNX1/RUNX1T1*	Favorable?	[50,59]
t(1;22)(p13;q13)	*RBM15-MKL1*	0.3%	0.7 (0.1–2.7)	Only M7 (5–10% of M7) Hepatosplenomegaly, fibrosis	Mainly no ACA, HD karyotypes		Intermediate	[48,60,61,62,63,64]
inv(16)(p13q24) **	*CBFA2T3-GLIS2*	2–3%	1.5 (0.3–17.2)	Infants, 20% of non-DS-AMKL, extramedullary disease, CD56++	Low HD karyotypes, tri 3, tri 21	Few mutations	Adverse	[46,48,50,64,65,66,67]
t(9;22)(q34;q11)	*BCR-ABL1*	0.6%		Exclude CML-BP or MPAL mBCR Sensitivity to TKI	Association with inv(16)/*CBFB-MYH11*		Adverse	[1,14,22]
UNBALANCED CA								
Monosomy 5, del(5q)	/	1.2%	12.5 (0.3–20.7)	M0	del(17p), CK		Adverse	[7,22,28,68]
Monosomy 7 ***	/	3%	7.2 (0–18)	Exclude a primary CA and a predisposition syndrome (*GATA2*)	/		Adverse	[22,28,69]
del(7q) ***	/	3%	7.6 (0–18)	Exclude a primary abnormality and a predisposition syndrome	/		intermediate	[22,28,69]
Trisomy 8 ***	/	10–14%	10.1 (0–18)	Mainly a secondary abnormality Search for a primary CA	/	*FLT3-ITD*	Discussed	[70]
Hyperdiploidy (48~49–65 chr.)	tri 8, tri 21, tri 19, tri 6, ….	11%	2 (0–17)	AMKL, infants, Search for a primary CA	/	/	No significance	[56,71]
Complex karyotype ƒ	/	8–17%	3 (0–18)	Exclude a primary CA	/	/	Discussed	[5,6,22,28]
Monosomal karyotype ƒƒ	/	3–5%	3.6 (0–17)	Exclude a CBF leukemia	/	/	Discussed/ Adverse even after exclusion of mon 7	[5,6]
Normal Karyotype								
Normal karyotype	/	20–26%	8.8 (0–18)	Search for a cryptic CA		Search for prognostic mutations: *FLT3-ITD,* *CEBPAdm, NPM1*	According to cryptic CA or to mutations	[7,22,28,36,46]

NOTE 1. Risk categories were defined according to Harrison [22] and Von Neuhoff 2010 [28]: Favorable, Intermediate and Adverse correspond to 5-year survival >70%, 50–70% and <50%, respectively. NOTE 2. Infants: children under 2 years. Abbreviations: APL: acute promyelocytic leukemia; CA: cytogenetic abnormality; CK: complex karyotype (at least 3 CAs); CML-BP: chronic myeloid leukemia blast phase; DIVC: disseminated intravascular coagulation; HD: hyperdiploid karyotype; mBCR: minor BCR; MPAL: mixed phenotype acute leukemia, mon: monosomy; r: rearrangement; TKI: tyrosine kinase inhibitors; tri: trisomy. * A complex rearrangement or a cryptic insertion is necessary to create a *KMT2A-MLLT10* fusion gene (see text); thus, FISH with a *KMT2A* probe is mandatory. ** Cryptic abnormality requiring molecular methods for detection: FISH and/or PCR-based method. *** As a primary abnormality. ƒ At least 3 independent CAs in the absence of a WHO-designated recurring translocation or inversion. Some authors include in the definition “with at least one structural abnormality” [5,28]. ƒƒ Loss of at least two autosomes or loss of one autosome and the presence of a structural abnormality (excluding mar or ring), excluding CBF AML.

## Data Availability

Not applicable.

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
