# Peer review of "Cytogenetics of Pediatric Acute Myeloid Leukemia: A Review of the Current Knowledge"

_genes, 2021, doi:10.3390/genes12060924_

Round 1

Reviewer 1 Report

This an important review in the time of more and more complex diagnostics at initial diagnosis.

I would recommend the publication; however, I suggest a careful review of typos and english language and editing of text/tables.

Main point: The technical part of this review is important, new and helpful for readers, highligting difficulties in techniques. The quality would be enhanced via inclusion of a graphical abstract/overview about your suggested workflow of cytogenetics at initial diagnosis

This may also include different situations (FAB types, age etc) and obligatory/suggested techniques. This could be also included in a short statement at the end of the main text.

Please find below my other comments:

Table 1:

In general, this table is helpful. However, an inclusion of details about the cytogenetic analyses (suggestions/pitfalls/specialties) in a separate column will enhance the quality and novelty of this review.

Other suggestions for Table 1:

  • Please include a table legend including also definitions for special characters such as * and **.
  • Please specify FLT3 in t(8;21). Do you mean FLT3-ITD? Or TKD?
  • Please specify/and correct FLT4(?) in inv 16
  • If possible, please specify in in the table the range for survival for the different CA or what mean “Good, intermediate or poor” in the table legend.
  • The percentages of KMT2Ar are included in the columns “special feature” but should be included in”Frequency” instead.
  • For consistency: t(7;12)(q36;p13)/ ETV6;MNX1 should be included in the table or moved to „rare translocations” in the main text.

APL:

  • You may add the information that APL constitutes an emergency at diagnosis “already with low WBC counts”. (line 162)
  • Please specify in your discussion about TP53 mut and complex karyotypes, if you describe observations of pediatric or adult APL (line 170 – line 182)
  • Please include the role of MRD monitoring in APL as you mention it for CBF leukemia.

CBF:

  • “These abnormalities have been confirmed and common deleted regions (CDR) refined by CGH or SNP array analyses, for example del(7)(q35q36.1) and del(9)(q21.2q21.3)”  (line 216/217). This sentence is not clear. Please correct the sentence.
  • However, del(9q) and trisomy 4 have been linked to a less favorable prognosis (line 220). This is true for the achievement of CR, but not linked to a reduced survival. Please specify/correct.

Inversion (3;3)(q21q26.2)/t(3;3)(q21;q26.2)/GATA2-MECOM (EVI1)

  • You may add a potential association with central diabetes insipidus as it can have clinical consequences.

Others:

  • Line 402 and 412: Specify “Survival” --> overall survival?

  • Line 419: OMS classification.? WHO?

  • Line 563-567: Study of the molecular landscape of pediatric AML confirms the association of well established cytogenetic subtypes with copy number alterations (CNA) and mutations, identifies novel fusion genes, unravels the complex interactions between these abnormalities, highlights age specific cytogenetic /molecular profiles and emphasizes the need for tailored/targetable therapies.This sentence is difficult to understand. Please correct.

  • Line 599: Finally, GATA1 mutations which are characteristic of DS-AMKL can be found too in 9% of non DS-AMKL and trisomy 21 is a constant feature in these cases. Are these cases DS mosaicism? Have they been analysed? Please specify in the text.

Table 2:

  • In the heading you specify that it is non-DS AMKL, but you include DS-AMKL in the table. This should be consistent.
  • The frequencies of KMT2Ar should be in the respective column and are no special features.

Conclusion: Please include a sentence in the conclusion, discussing the need of cytogenetics in the time of emerging methods (Sequencing etc).

The supplemental Figure is the same as Figure 2?

 Just a few examples for typos:

-abtract  

- line 157: APL is represents

- line 310, 311 and 318 “)

- line 508: prognosis factors 

- Line 631: inv(16/CBF-MYH11

Author Response

Please see atttachement containing responses to both Reviewers

Reviewer 2 Report

The review is very thorough and updated. Would be good if it is mentioned and commented more about the difference and similarities in risk-classification between pediatric and adult AML, maybe as additional Table. 

On Table 1 it is written FLT4 with inv(16). It is most likely FLT3. Page 10, line 418, it is written OMS classification?! There are more typos in the text.

Suggestion: update text with new information recently published in Blood Advances about the effect on prognosis with ACA in adult CBF AML: https://doi.org/10.1182/bloodadvances.2020003605

Another suggestion is to mention difficulties in obtaining metaphases and value of unsuccessful cytogenetics and new methods which can overcome this obstacle as:

  • 10.1111/ejh.12446
  • 10.1056/NEJMoa2024534

Author Response

Please find the attachement containing responses to both Reviewers as a detailed Cover letter
